# Effect of Microwave Sintering on the Properties of 0.95(Ca_0.88_Sr_0.12_)TiO_3_–0.05(Bi_0.5_Na_0.5_)TiO_3_ Ceramics

**DOI:** 10.3390/ma12050803

**Published:** 2019-03-08

**Authors:** Lei Tian, Jing Nan, He Wang, Chunying Shen

**Affiliations:** 1College of Materials Science and Engineering, Nanjing Tech University, Nanjing 210009, China; tianlei0206@foxmail.com (L.T.); nanjing2016@njtech.edu.cn (J.N.); wanghe0213@njtech.edu.cn (H.W.); 2Jiangsu Collaborative Innovation Center for Advanced Inorganic Function Composites, Nanjing Tech University, Nanjing 210009, China

**Keywords:** dielectric properties, microwave sintering, temperature coefficient, ceramics

## Abstract

Perovskite ceramics are a common microwave dielectric material, but the development and application of this material has been limited by the high, positive resonance frequency temperature coefficient and sintering temperature. Therefore, adjusting the temperature coefficient of the resonance frequency and reducing the sintering temperature have become important research directions. In this work, 0.95(Ca_0.88_Sr_0.12_)TiO_3_–0.05(Bi_0.5_Na_0.5_)TiO_3_ ceramics (referred to as 0.95CST-0.05BNT) were prepared by standard solid-state reaction and microwave sintering. Microwave sintering greatly shortened the sintering period and holding time. Moreover, the 0.95CST–0.05BNT ceramics showed more uniform grain size distribution, and microwave sintering reduced energy consumption in the experiment. Therefore, the temperature coefficient of the resonance frequency of MWS ceramics was reduced by 119 × 10^−6^ /℃. All of the ceramics, which were sintered at 1300 °C for 40 minutes, showed optimal microwave dielectric properties: *ε_r_* = 187.6, *Q* × *f* = 8958 GHz, and *τ_f_* = +520 × 10^−6^ /°C.

## 1. Introduction

With the rapid development of wireless communication technology, microwave dielectric materials have been extensively used in many microwave-integrated circuits, such as dielectric oscillators, filters, and other devices [1]. Generally, microwave dielectric ceramics require three key properties, including dielectric constant (*ε_r_*), a high *Q × f* value, and a low-temperature coefficient of resonance frequency (*τ_f_*) to maintain the temperature stability of microwave ceramics [2,3]. Owing to their high dielectric properties, the microwave ceramics have been in high demand in recent years.

Recently, many perovskite ceramics have attracted much attention due to their good dielectric properties and the adjustable nature of the perovskite structure [4]. Examples include Ca_0.6_Sr_0.4_TiO_3_ ceramics (*ε_r_* = 218, *Q × f* = 7182 GHz, *τ_f_* = +1164 × 10^−6^ /°C), CaTiO_3_ ceramics (*ε_r_* = 174, *Q × f* = 6005 GHz, *τ_f_* = +824 × 10^−6^ /°C) [5], and (Ca_0.2_Sr_0.8_)_3_Ti_2_O_7_ ceramics (*ε_r_* = 58, *Q × f* = 25727 GHz, *τ_f_* = +359 × 10^−6^ /°C) [6]. (Ca_0.88_Sr_0.12_)TiO_3_ (*ε_r_* = 173, *Q × f* = 8310 GHz, *τ_f_* = +942 × 10^−6^ /°C) ceramics which, have a large positive *τ_f_* value, have been widely studied; meanwhile, (Na_0.5_Bi_0.5_)TiO_3_, which has a negative *τ_f_* value of –180×10^−6^ /°C and a higher dielectric constant of 480, is often used as an auxiliary material [7,8,9]. Thus, (Na_0.5_Bi_0.5_)TiO_3_ is added to (Ca_0.88_Sr_0.12_)TiO_3_ to compensate for its *τ_f_*. In our previous work, (1–x)(Ca_0.88_Sr_0.12_)TiO_3_-x(Bi_0.5_Na_0.5_)TiO_3_ ceramics were prepared by the standard solid-state sintering method. The 0.95CST–0.05BNT ceramics, having excellent performance (*ε_r_* = 185.5, *Q × f* = 8116 GHz, *τ_f_* = + 639 × 10^−6^ /°C), had been obtained [10].

Nowadays, the microwave sintering of ceramic materials is a new type of sintering process both at home and abroad [11,12]. The basic heating principle of microwave sintering is quite different from that of conventional sintering. During the microwave heating, the polarization occurs inside the material, which makes the dipole turn repeatedly with the fast-changing alternating electromagnetic field. Since the frequency of the magnetic field changes very fast, the vibration and friction between the dipoles in the material become more intense; thus, the material itself heats up and warms up to a certain temperature. Compared with conventional sintering methods, we summarize the characteristics of several microwave sintering methods as follows. 1. Due the effect of the electromagnetic field, the material is heated by itself. The overall heating of the material is more uniform; in addition, the heating method avoids the additional internal stress produced by uneven heating, and reduces the internal defects of the materials, which reduces the material loss and makes the material difficult to crack due to fast speed of the heating process. 2. Microwave sintering can greatly shorten the sintering cycle, increase the sintering rate, and reduce the sintering temperature to obtain ceramics with high densification. At the same time, microwave sintering can optimize the properties of materials. 3. Microwave sintering can optimize the microstructures of materials. Since the heating rate is very fast during microwave sintering, it can effectively inhibit grain growth, which can successfully prepare a nanometer-level superfine powder.

In this paper, 0.95CST–0.05BNT ceramics were prepared by standard solid-state and microwave sintering. The microwave sintering method has the advantages of a short sintering period and low energy consumption. The effect of microwave sintering on microstructures and properties were presented and discussed. Finally, we summarized the differences between microwave sintering (MWS) and conventional sintering (CS) in a table. Microwave sintering greatly shortened the sintering period and reduced the energy consumption.

## 2. Experimental

AR raw powders of CaCO_3_ (Shanghai Aladdin Biochemical Technology Co., Ltd. Shanghai, China), SrCO_3_ (China Pharmaceutical Group Chemical Reagents Co., Ltd. Shanghai, China), Bi_2_O_3_ (Xilong Science Co., Ltd. Shantou, China), Na_2_CO_3_ (Yixing Chemical Reagent Factory, Yixing, China), and TiO_2_ (Xiantao Zhongxing Electronic Materials Co., Ltd. Xiantao, China) (>99%) were weighed according to the stoichiometric ratio of (Ca_0.88_Sr_0.12_)TiO_3_ and (Bi_0.5_Na_0.5_)TiO_3_. The (Ca_0.88_Sr_0.12_)TiO_3_ powders were milled for six hours with ZrO_2_ balls and deionized water. The (Bi_0.5_Na_0.5_)TiO_3_ powders were milled for six hours with ZrO_2_ balls and ethanol. The ball-milling parameters were as follows: power-to-ball weight ratio: 1:2; powder-to-water weight ratio: 1:1.5; powder-to-ethanol weight ratio: 1:1.2. Afterwards, the mixture was dried and calcined at 1100 °C and 850 °C for three hours with heating and cooling rates of 5 °C/min, respectively. Then, the calcined powders were weighed according to the formula of 0.95CST–0.05BNT, and the powders were re-milled for six hours by ball milling in water media at the same time. The mixture with PVA solution was pressed into cylinders under a uniaxial pressure of 300 MPa, and sintered at 1275–1350 °C for 20 to 50 minutes by microwave sintering. For a comparison, 0.95CST–0.05BNT ceramics were also prepared at 1275 °C for three hours by conventional sintering. The microwave furnace (MW-L0316V, Changsha Longtech Co. Ltd., Changsha, China) consisted of 2.45-GHz magnetrons with a maximum power of three kW.

The phase constitution was identified by powder X-ray diffraction using CuKα radiation with a scan width of 10–80° a and scan speed of 10 °/min (XRD; RIGAKU; SmartLab 3; 40kV; 30mA, Beijing, China). The microstructures were observed on the surfaces with scanning electron microscopy (SEM; JEOL, JSM-5900; 15kV, Beijing, China). The sample density was determined by the Archimedes method. The cylinders with a diameter of about 15 mm and thickness of about 7.3 mm were used for evaluating the microwave dielectric properties using a vector network analyzer (Agilent 8722ET) [13,14].

## 3. Results and Discussion

Figure 1 exhibits the XRD patterns of 0.95CST–0.05BNT ceramics sintered by CS and MWS. We could see that all the samples displayed the characteristic peaks of perovskite-type structures, which could be indexed to be the perovskite structure (ICSD-PDF#81-0562), and no other phase was identified. It showed that different sintering methods had little effect on phase composition.

The SEM micrographs on the surfaces of 0.95CST–0.05BNT samples sintered at 1300 °C for different holding times by MWS are shown in Figure 2a–d. The average grain size increased, and the number of pores reduced gradually as the holding time increased. In order to further compare the difference between MWS and CS ceramics, Figure 2e illustrates the SEM of 0.95CST–0.05BNT samples prepared by conventional sintering at 1275 °C for three hours. It could be seen in Figure 2c, e that there was little change in the grain shape of the MWS samples, but the grain size of the MWS samples (one μm) was smaller than that of the CS samples (three μm). It could be explained that the sintering time was much shorter and the heating rates were much faster in the process of microwaving heating than that of the conventional heating; thus, fine and uniform grains were obtained.

The bulk density of 0.95CST–0.05BNT ceramics as a function of different sintering temperatures and holding times is shown in Figure 3. The bulk density increased gradually as the holding time increased. The highest density was obtained when the sintering temperature was 1300 °C for 40 minutes. However, the bulk densities did not increase significantly when the sintering temperature exceeded the optimal value. The grain grew completely in the ultra-high sintering temperature, and the densities of the ceramics were also optimal. All of the samples showed that the optimum bulk densities were obtained when the sintering temperature was 1300 °C for 40 minutes. 

Figure 4 and Figure 5 show the dielectric constant (*ε_r_*) and quality factor (*Q × f*) of 0.95CST–0.05BNT samples as functions of different sintering temperatures and holding times. It could be seen from the diagram that the dielectric properties of the samples had been greatly improved with the increased temperature and holding time. The dielectric properties first increased, and then stabilized as the holding time increased. The dielectric constant and quality factor obtained a maximal value when the samples were sintered at above 1300 °C. Considering the sintering and dielectric properties, energy consumption, and other factors, the optimum conditions for the preparation of ceramics by microwave sintering were as follows: the sintering temperature and holding time were 1300 °C and 40 minutes, respectively.

Figure 6 shows the time–temperature curve for 0.95CST–0.05BNT ceramics sintered by MWS and CS. The time required for microwave sintering was much shorter than that of conventional sintering. The sintering period was just about six hours, and the sintering temperature up to 1300 °C only needed 58 minutes by microwave sintering. Moreover, the holding time was only 40 minutes. In contrast, conventional sintering required at least 12.75 hours to reach the highest temperature, and its holding time extended to three h, which led to a sintering period of at least 30 hours.

Table 1 summarizes the differences between conventional sintering and microwave sintering on the properties of 0.95CST–0.05BNT samples. As we could see, microwave sintering not only shortened the sintering period and holding time, but also improved the properties. The dielectric constant of the MWS samples were higher than those of the CS samples. Overall whole, it was the change of the dielectric constant that was closely consistent with the density of the ceramics. The main cause accounting for the increasing of the dielectric constant was due to the enhancement of the densification. From Table 1, it could be also concluded that the quality factor of the MWS samples was higher than that of the CS samples. This phenomenon was attributed to the more uniform grain size distribution and compact microstructure of the MWS samples. Under microwave radiation, the movement of vacancies and interstitial ions, which parallel the electric field, enhanced instantaneously, which greatly reduced the diffusion barrier, permitting ions to move faster and expediting the grain boundary diffusion and densification rates [15,16,17]. However, during the conventional sintering, heat was transmitted between objects through conduction, radiation, and convection mechanisms. In addition, conventional sintering led to temperature gradients from the surface to the inside. Therefore, microwave sintering resulted in samples with a more uniform grain size distribution and densification. Besides, the temperature coefficient of resonance frequency (*τ_f_*) of the MWS samples was also closer to zero, which decreased greatly from +639 to +520 × 10^−6^ /°C. 

## 4. Conclusions

The 0.95CST–0.05BNT samples had been prepared successfully by microwave sintering. The sintering period only needed six hours by microwave sintering; thus, this method could greatly improve production efficiency. Moreover, the 0.95CST–0.05BNT ceramics showed more uniform grain size distribution, and microwave sintering greatly shortened the sintering period and holding time. As a result, the optimal microwave dielectric properties with a dielectric constant of 187.6, *Q × f* value of 8958 GHz, and temperature coefficient of resonant frequency of +520 × 10^−6^ /°C were achieved at 1300 °C for 40 minutes by microwave sintering. In particular, the temperature coefficient of resonance frequency (*τ_f_*) of MWS ceramics was also closer to zero, which decreased greatly from +639 to +520 × 10^−6^ /°C.

## Figures and Tables

**Figure 1 materials-12-00803-f001:**
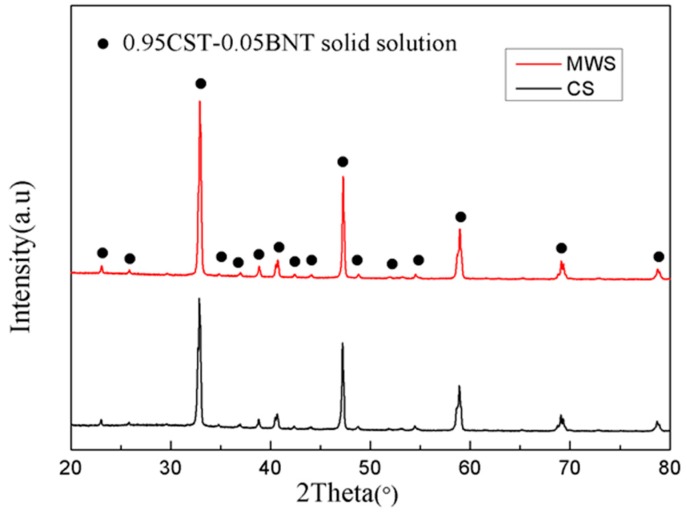
XRD patterns of 0.95CST–0.05BNT samples sintered by different methods.

**Figure 2 materials-12-00803-f002:**
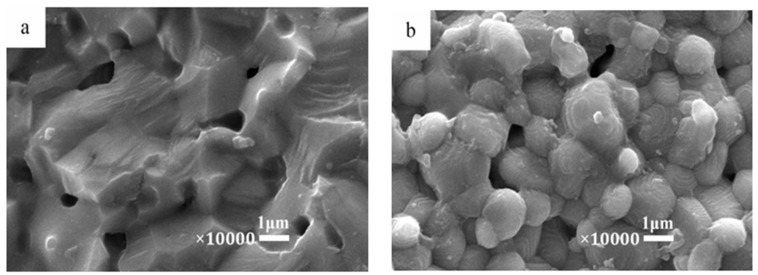
SEM micrographs of 0.95CST–0.05BNT samples sintering at 1300 °C by microwave sintering (MWS), with different holding times: (**a**) 20 min, (**b**) 30 min, (**c**) 40 min, (**d**) 50 min; (**e**) 1275 °C hold for three hours by conventional sintering (CS).

**Figure 3 materials-12-00803-f003:**
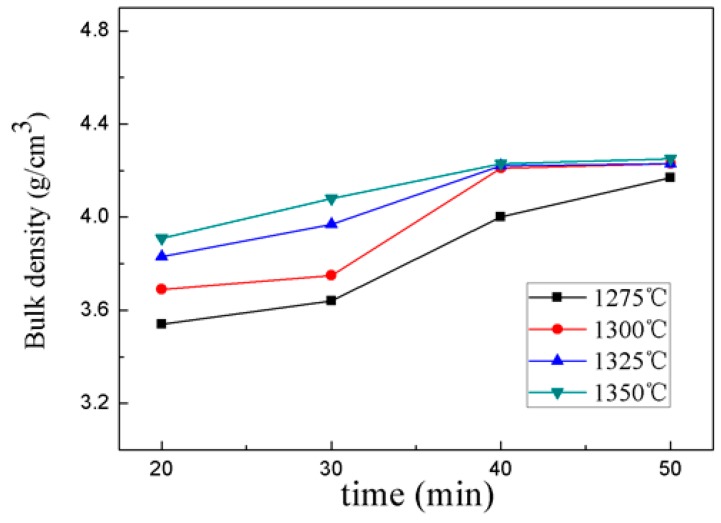
Bulk density of the 0.95CST–0.05BNT samples with different sintering temperatures and holding times by MWS.

**Figure 4 materials-12-00803-f004:**
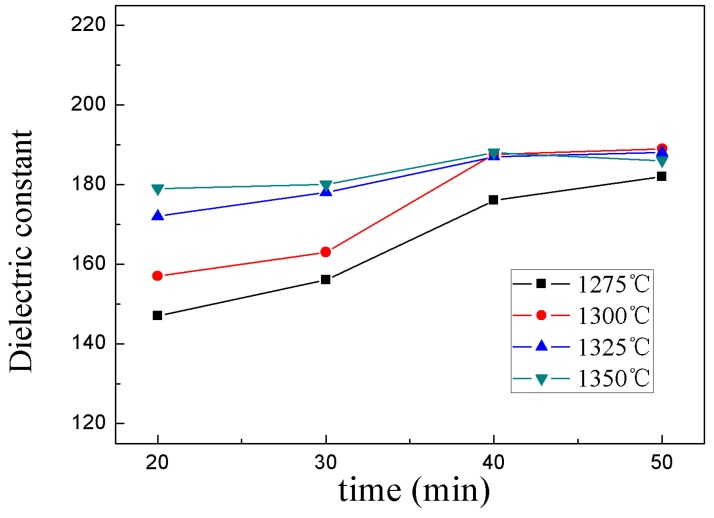
Dielectric constant *ε_r_* of the 0.95CST–0.05BNT samples with different sintering temperatures and holding times by MWS.

**Figure 5 materials-12-00803-f005:**
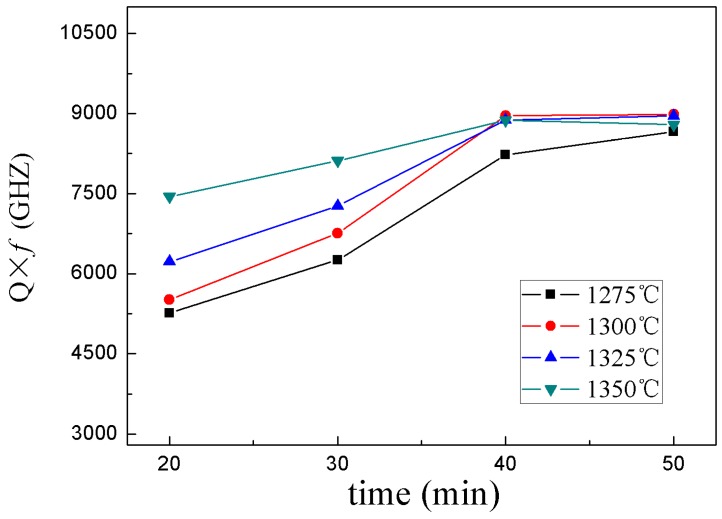
*Q × f* value of the 0.95CST–0.05BNT samples with different sintering temperatures and holding times by MWS.

**Figure 6 materials-12-00803-f006:**
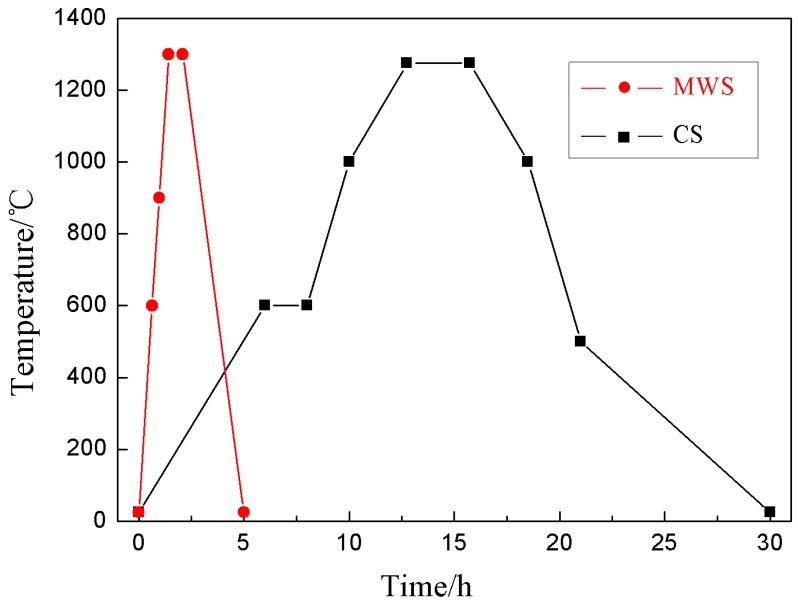
Time–temperature sintering curve for MWS and CS.

**Table 1 materials-12-00803-t001:** Microwave properties of 0.95CST–0.05BNT samples prepared with different sintering processes.

Parameters	MWS (CST-BNT)	CS (CST-BNT)
Sintering period (h)	6	30
Holding time (h)	0.67	3
Sintering temperature (°C)	1300	1275
Bulk densities (g/cm^3^)	4.23	4.22
Dielectric constant	187.6	185.5
Quality factor (GHz)	8958	8116
*τ_f_* (×10^−6^ /℃)	+520	+639

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
