# Peer review of "Effect of Microwave Sintering on the Properties of 0.95(Ca_0.88_Sr_0.12_)TiO_3_–0.05(Bi_0.5_Na_0.5_)TiO_3_ Ceramics"

_materials, 2019, doi:10.3390/ma12050803_

Reviewer 1 Report

Dear Sir,
Based on revision of the paper I would like to recommend the following:
In the abstract section

There should be a general sentence to describe neither the method nor the materials under investigation or the general topic which grasp the attention of the researcher or the scientific community (there is no kind of that).

Thus, the abstract should be rephrased including the regular shape of similar work in this field.

Additionally, it is recommended to use short keywords for each one e.g. Temperature coefficient of resonance frequency is too long.

Introduction section

·        Some grammatical error should be corrected sand there is some typos in the manuscript for example :

-         pay attention to the ceramics should be: pay attention to that ceramics

-         such as CaTiO3 ceramics should be Such as CaTiO3 ceramics (capital letter)

-         Space should be left between the number and its unit of measurement.

-         It is better to use thus instead of so

The experimental part

-         Grade and company of the supplied chemicals should be stated.

-         the model and country of origin of the instrument should be stated such as balance, water purification system, muffle furnace, etc.

-         ball-to-power weight ratio: 2 not clear should be 2:1

-         state the heating and cooling rate of the first calcination step.

-         More detail should be stated about XRD parameters such as scan width and scan speed, etc.

The result and discussion

More details should be stated in the XRD measurements.

Better resolution is needed for Fig 2. A) and fig 2. B).

so other characterization can be done for the obtained materials such as TEM and BET to check the surface of the obtained materials.

Conclusion part

In conclusion should be removed and especially should be replaced by in particular.

   Reference

 should be checked carefully according to the style of the journal for some references.

 Regards

Author Response

Response to Reviewer 1 Comments

Point 1: In the abstract section; Introduction section; The experimental part.

Response 1: All your comments have been revised.

Point 2: The result and discussion

Response 2:  Fig 2a and fig 2b use better resolution. In SEM and tabel, I do a more detailed analysis for sintering and dielectric properties. As suggested by the reviewers, we have change the description " It could be seen in Fig.2c, e that the grain shape of the MWS samples changed little, but the grain size of the MWS samples (1 μm) was smaller than that of the CS samples(3 μm). This is due to the fact that the sintering time is much shorter and the heating rates is much faster in the process of microwaving heating than that of the conventional heating, thus fine and uniform grains were obtained." " Table 1 summarizes the differences between conventional sintering and microwave sintering on sintering and dielectric properties of 0.95CST-0.05BNT ceramics. As we could see, microwave sintering not only shortened the sintering period and holding time, but also improved the sintering and dielectric properties. The dielectric constant of the microwave specimen is higher than that of conventional one. In general, the change of the dielectric constant is closely consistent with the density of the materials. The reason for the increasing of dielectric constant was owing to the decrease of pores and the enhancement of the densification. We can also observe from the table that the quality factor of microwave sintering samples was higher than that of conventional sintering samples. This behavior was attributed to the more uniform grain growth and compact microstructure for the microwave-sintered products. Under the radiation of microwave, the materials are coupled with the electromagnetic fields to absorb the microwave energy, and then convert it into heat directly. While, in the conventional process, heat is transferred between objects by the mechanisms of conduction, radiation ad convection. Furthermore, the conventional process causes the appearance of the temperature gradient from the surface to the inside. Hence, the heating mechanism of microwave could lead to the samples with more uniform grain distribution and densification. Compared with the conventional sintering, the temperature coefficient of resonance frequency (τf) of MWS ceramics was also closer to zero, which decreased greatly from +639 to +520×10-6 /°C. "

Point 3:  Reference

Response 3: As suggested by the reviewers, we have change the description.

Reviewer 2 Report

ABSTRACT

Should be shortened and more focussed

In particular rephrasing is necessary

INTRODUCTION

The first two paragraphs (1.5 pages) are too general with respect to a great variety of materials treatable by microwave; please set a focus! Give explicit values for sintering of example materials (What is the sintering time and temperature for alumina under conventional conditions?)

What does this sentence mean: “Recently, because of perovskite ceramics have the adjustable of perovskite structure and excellent dielectric properties,….”

Please convert ppm/°C in SI units

Pls. extract results given at the end of chapter INTRODUCTION (e. g. “….The microwave sintering method has the advantages of short sintering period and low energy consumption…..”) this is pat of the discussion

EXPERIMENTAL

Pls. give name, affiliation etc. of material´s suppliers as it s standard in publications

Vial materialàball mill lining material (vial confuses)

Ball-to-power or ball-to-powder ratio?

Water-to-powder volume ratio?

What is CST? What is BNT? Pls. no abbreviation in the abstract; define properly!

What is appropriate polyvinylalcohol? Pls. give amount, ratio

Pls. give data concerning the microwave device used in this work (type, supplier, power data applied)

Tube current and voltage of XRD, SEM devices?

RESULTS AND DISCUSSION

Pls. label all reflexes (not peaks) to identify crystallographic planes

Question: The reflex shape and intensity ratios seem to be the same for CS compared to MWS; how can you derive the statement “….the intensities of MWS ceramics diffraction peak were stronger than that of CS ceramics, which illustrated that the ceramics have better crystallinity by microwave sintering….”; please justify! If possible: pls. perform a Rietveld analysis and give more specific data on the structure

Figure captions: pls. name the methods and sintering conditions also here.

What is pore discharging? Pls. explain! What is meant when stating “the density is stabilized?

What sintering properties have been improved by microwave sintering? Isn´t it the microstructure which has changed to the positive?

CTE of the resonance frequency: Your statement says this is by a reduced Bi evaporation. Question: Is there any evidence for this, e.g. by elemental analysis? This is easy to measure, otherwise it is speculation.

CONCLUSIONS

A more homogeneous grain size distribution is stated. Can this be given as a value and such a value (half life with and maximum of a grain size distribution, for instance) compared to that of CS materials?

Author Response

Response to Reviewer 2 Comments

Point 1: In the abstract section; Introduction section; The experimental part.

Response 1: All your comments have been revised.

Point 2: The reflex shape and intensity ratios seem to be the same for CS compared to MWS; how can you derive the statement “….the intensities of MWS ceramics diffraction peak were stronger than that of CS ceramics, which illustrated that the ceramics have better crystallinity by microwave sintering….”; please justify! If possible: pls. perform a Rietveld analysis and give more specific data on the structure.

Response 2: As suggested by the reviewers, we have change the description "the XRD patterns of 0.95CST-0.05BNT ceramics sintered by CS and MWS at the best craft conditions. What we could learn from the patterns is that all the samples displayed the characteristic peaks of perovskite-type structures, which can be indexed to be the perovskite structure (ICSD-PDF#81-0562) and no other phase was identified. It shows that different sintering methods have little effect on phase composition."

Point 3: What is pore discharging? Pls. explain! What is meant when stating “the density is stabilized?

Response 3: As suggested by the reviewers, we have change the description " It could be explained that ultra-high sintering temperature resulted in the grain growing complete and densities of ceramics were also optimal."

Point 4: What sintering properties have been improved by microwave sintering? Isn´t it the microstructure which has changed to the positive?

Response 4: As suggested by the reviewers, we have change the description " It could be seen in Fig.2c, e that the grain shape of the MWS samples changed little, but the grain size of the MWS samples (1 μm) was smaller than that of the CS samples(3 μm). This is due to the fact that the sintering time is much shorter and the heating rates is much faster in the process of microwaving heating than that of the conventional heating, thus fine and uniform grains were obtained."

Point 5: CTE of the resonance frequency: Your statement says this is by a reduced Bi evaporation. Question: Is there any evidence for this, e.g. by elemental analysis? This is easy to measure, otherwise it is speculation.

Response 5: As suggested by the reviewers, we have change the description "We can also observe from the table that the quality factor of microwave sintering samples was higher than that of conventional sintering samples. This behavior was attributed to the more uniform grain growth and compact microstructure for the microwave-sintered products. Under the radiation of microwave, the materials are coupled with the electromagnetic fields to absorb the microwave energy, and then convert it into heat directly. While, in the conventional process, heat is transferred between objects by the mechanisms of conduction, radiation ad convection. Furthermore, the conventional process causes the appearance of the temperature gradient from the surface to the inside. Hence, the heating mechanism of microwave could lead to the samples with more uniform grain distribution and densification. Compared with the conventional sintering, the temperature coefficient of resonance frequency (τf) of MWS ceramics was also closer to zero, which decreased greatly from +639 to +520×10-6 /°C."

Point 6: A more homogeneous grain size distribution is stated. Can this be given as a value and such a value (half life with and maximum of a grain size distribution, for instance) compared to that of CS materials?

Response 6: As suggested by the reviewers, we have change the description " the grain size of the MWS samples (1 μm) was smaller than that of the CS samples(3 μm)."

Round  2

Reviewer 1 Report

Thank you for your efforts to improve the manuscript.